# Barriers and facilitators to vaccination uptake against COVID-19, influenza, and pneumococcal pneumonia in immunosuppressed adults with immune-mediated inflammatory diseases: A qualitative interview study during the COVID-19 pandemic

**Amy Fuller[1]\*, Jennie Hancox[2], Kavita Vedhara[2], Tim Card[3], Christian Mallen[4], Jonathan S. Nguyen Van-Tam[3], Abhishek Abhishek[1]**

1 Academic Rheumatology, School of Medicine, University of Nottingham, Nottingham, United Kingdom, 2 Centre for Academic Primary Care, Lifespan and Population Health, School of Medicine, University of Nottingham, Nottingham, United Kingdom, 3 Lifespan and Population Health, School of Medicine, University of Nottingham, Nottingham, United Kingdom, 4 School of Medicine, Keele University, Keele, United Kingdom

\* amy.fuller@nottingham.ac.uk

## Abstract

### Objectives

To explore barriers and facilitators to COVID-19, influenza, and pneumococcal vaccine uptake in immunosuppressed adults with immune-mediated inflammatory diseases (IMIDs).

### Methods

Recruiting through national patient charities and a local hospital, participants were invited to take part in an in-depth, one-to-one, semi-structured interview with a trained qualitative researcher between November 2021 and January 2022. Data were analysed thematically in NVivo, cross-validated by a second coder and mapped to the SAGE vaccine hesitancy matrix.

### Results

Twenty participants (75% female, 20% non-white) were recruited. Barriers and facilitators spanned contextual, individual/group and vaccine/vaccination-specific factors. Key facilitators to all vaccines were higher perceived infection risk and belief that vaccination is beneficial. Key barriers to all vaccines were belief that vaccination could trigger IMID flare, and active IMID. Key facilitators specific to COVID-19 vaccines included media focus, high incidence, mass-vaccination programme with visible impact, social responsibility, and healthcare professionals' (HCP) confirmation of the new vaccines' suitability for their IMID. Novel vaccine technology was a concern, not a barrier. Key facilitators of influenza/pneumococcal

**Data Availability Statement:** Transcript excerpts relevant to the study are included within the paper, and the interview guide has been included as a Supporting Information file. Additional qualitative data and audio recordings are archived in the University of Nottingham servers using password protection, and are restricted due to privacy concerns. Data are available upon request from the Research Governance team (sponsor@nottingham.ac.uk) for researchers who meet the criteria for access to confidential data.

**Funding:** This project is funded by the National Institute for Health Research (NIHR) under its Research for Patient Benefit (RfPB) Programme (Grant Reference Number NIHR 201973). The views expressed are those of the author(s) and not necessarily those of the NIHR or the Department of Health and Social Care. The funders had no role in study design, data collection and analysis, decision to publish, or preparation of the manuscript.

**Competing interests:** I have read the journal's policy and the authors of this manuscript have the following competing interests: AF, JH, KV, TC and JSN-V-T declare no competing interests. CM declares grants awarded from MRC, AHRC BMS and Versus Arthritis, and is Director of the NIHR SPCR. AA declares grants from AstraZeneca and Oxford Immunotec, personal payments from UpToDate, Springer, Menarini and Cadilla pharmaceuticals, consulting fees from Inflazome and NGM Biopharmaceuticals, meeting attendance/travel payments from Pfizer, and is co-chair for the OMERACT CPPD classification criteria and ACR/EULAR CPPD classification criteria working groups. JSN-V-T was seconded to the Department of Health and Social Care, England (DHSC) until 31st March 2022. The views expressed in this manuscript are those of its authors and not necessarily those of DHSC or the Joint Committee on Vaccination and Immunisation (JCVI). These do not alter our adherence to PLOS ONE policies on sharing data and materials.

vaccines were awareness of eligibility, direct invitation, and, clear recommendation from trusted HCP. Key barriers of influenza/pneumococcal vaccines were unaware of eligibility, no direct invitation or recommendation from HCP, low perceived infection risk, and no perceived benefit from vaccination.

## Conclusions

Numerous barriers and facilitators to vaccination, varying by vaccine-type, exist for immunosuppressed-IMID patients. Addressing vaccine benefits and safety for IMID-patients in clinical practice, direct invitation, and public-health messaging highlighting immunosuppression as key vaccination-eligibility criteria may optimise uptake, although further research should assess this.

## Introduction

Immune-mediated inflammatory diseases (IMID) such as rheumatoid arthritis, inflammatory bowel disease, vasculitis and systemic lupus erythematosus affect around 1 in 50 adults [1–4]. People with these conditions have approximately 30% increased risk of hospitalisation and death from COVID-19 [5,6], 80% higher risk of influenza complications [7,8], as well as a higher risk of complications from pneumococcal pneumonia [9,10]. Despite long-standing recommendations [11], influenza and pneumococcal vaccine uptake among immunosuppressed-IMID patients remains sub-optimal [12,13], ranging between 38.2% for pneumococcal and 69.4% for influenza vaccine. Uptake is particularly low for under 65s and those without comorbidities that qualify them for invitation for vaccination [12,13]. People with IMID are also less likely to have been vaccinated against COVID-19 compared to those with non-inflammatory rheumatic diseases [14] with 37–46% IMID patients uncertain or unwilling to get vaccinated [15,16]. As effective drug treatments for COVID-19 are found, and less-serious SARS-CoV-2 variants emerge, the perceived benefit from booster doses may diminish. Thus, it is possible that the uptake of further vaccinations against COVID-19 may be lower in this patient cohort [17]. To the best of our knowledge, an in-depth qualitative study exploring barriers and enablers to COVID-19 vaccine uptake in IMID has not yet been conducted. Thus, the aim of our study was to identify barriers and facilitators to vaccine uptake against COVID-19, influenza and pneumococcal disease in immunosuppressed adults with a broad range of IMIDs. Several respiratory infections were included as they are vaccine-preventable illnesses and vaccines against influenza and COVID-19 will likely be administered together in the future.

## Methods

### Study design

In-depth qualitative interview study.

### Ethical approval

London–Brighton & Sussex Research Ethics Committee (21/PR/1147). The study was registered on clinicaltrials.gov (NCT05115370). Participants gave their informed consent via an online consent form prior to the interview.

### Eligibility and recruitment

IMID diagnosed adults aged > = 18 years and prescribed immune-suppressing drugs were eligible. Participants were recruited from national patient charities and outpatient clinics of Nottingham University Hospitals NHS Trust between November 2021 and January 2022. See S1 Supplementary methods for further details.

A combination of purposive stratified and maximum variation sampling was employed to recruit a mix of participants with:

- Different inflammatory conditions.

- Different levels of engagement with vaccination:

  ○ Always vaccinated: received all influenza vaccines over previous three years (or time since diagnosis if <3 years), and the pneumococcal vaccine;

  ○ Sometimes vaccinated: missed up to two of the above vaccine doses;

  ○ Often not vaccinated: missed three or more of the above vaccine doses.

- Presence or absence of comorbidities that qualify for vaccination against respiratory infections in the UK such as age > = 65 years, diabetes, asthma, etc. [18].

### Interviews

All interviews were conducted remotely by AF (Research Fellow, trained in qualitative interviewing) by telephone in a private room and digitally audio-recorded. To reduce the risk of response bias, the researcher explained they were not medically trained and would remain impartial to participants' decision to be vaccinated or not. Interviews followed a semi-structured guide (see S2 Supplementary methods) exploring participants' understanding of influenza, pneumonia and COVID-19, the risk these infections pose to their health, understanding of vaccinations for these infections and how they work; reasons for choosing to be vaccinated or not, and, whether the COVID-19 pandemic impacted their perception and/or engagement with vaccination.

### Patient and public involvement (PPI)

Three PPI members with an IMID and experience taking immunosuppressing medication took part in pilot interviews to test and review the interview guide. One provided input into participant-facing documents.

### Analysis

Interviews were transcribed verbatim using an external transcription service. Transcripts were checked for accuracy, anonymised and imported into NVivo 12. Data were thematically analysed [19] using a combined inductive and deductive approach. After four interviews, AF and JH independently read and coded segments of text. Initial ideas were discussed between the two researchers with good agreement. At this point it was recognised that the data mapped to the SAGE vaccine hesitancy matrix [20], which was employed as the working analytical framework, whilst allowing any new barriers and facilitators specific to the participants' experience of being immunosuppressed and/or their IMID to emerge from the data inductively. These were grouped into the wider themes as appropriate. Previously coded transcripts were checked for retrospective fit of new codes emerging during later analysis. Data was examined for potential differences between participants according to their IMID and medication type. Analysis

was conducted concurrently with data collection to check for saturation, which ceased once thematic data saturation was reached and a suitable mix of participants had been interviewed.

## Results

Twenty participants (six 'always', seven 'sometimes' and seven 'often not' vaccinated) were interviewed between November-2021 and January-2022 (S1 Table for participant characteristics). Interviews lasted 59 minutes on average (range 42–81). There were no differences between participants' responses according to their condition or medication. Barriers and facilitators to vaccination, by vaccine type, mapped to the SAGE working group matrix [20] (Table 1). A diagrammatic summary of the barriers and facilitators is presented in Fig 1.

**Table 1. Barriers and facilitators to vaccination among immunosuppressed adults with immune-mediated inflammatory diseases (IMID) by vaccine type, mapped to the SAGE working group matrix.**

|  | Facilitators | Barriers |
|---|---|---|
| **Contextual influences** | | |
| Communication and media | News on threat and seriousness of infection, importance and impact of vaccination on hospitalisation and deaths (C). Advertisement (F). | No news on importance of vaccination (F, P). Little to no advertising (P), or no mention of immunosuppressed being eligible (F, P). |
| Influential leaders | Encouragement from government, HCPs, notable scientists (C). Endorsement by patient organisations (C, F, P). | None |
| Covid-19 pandemic restrictions [a] | To return to normal (C). Categorised as Clinically Extremely Vulnerable (C, F, P). | None |
| **Individual and group influences** | | |
| Personal, family or community experience | Positive experience with previous vaccination (C, F, P). | Negative reports from friends and family (F) |
| Risk/benefit (perceived, heuristic) | High incidence of infection and few treatments (C). Higher risk of infection and complications (C, F, P). Confirmation by HCP of no interaction with IMID or its treatment (C, F, P). | Low perceived risk of infection (F, P). Belief that vaccination can or did trigger IMID flare up (C, F, P). |
| Beliefs, attitudes about health and prevention | Belief in benefit from vaccination (C, F, P). Social responsibility to protect others and minimise NHS burden [a] (C). Often in environment with high chance of catching infection (F, P). | Belief of no tangible benefit from vaccination (F, P). Shielding or in environments with low chance of catching infection (F, P). |
| Health system and providers— trust and personal experience | Trusted HCP familiar with IMID recommending vaccination (C, F, P) | |
| Immunisation as social norm | Has always accepted vaccinations (C, F, P) | None |
| Knowledge, awareness | Aware of eligibility (F, P). | Unaware of eligibility (F, P). |
| Priority [a] | None | Other medical appointments for IMID lowers priority of vaccination (F, P). |
| Disease state [a] | Stable IMID (C, F, P) | Unstable IMID (C, F, P) |
| **Vaccine or vaccination specific issues** | | |
| Design of vaccination programme or mode of delivery | Mass vaccination programme with ample appointments, prompts and HCP enquiry on vaccination status (C). Invitation from GP (F, P). Ease of booking (F, P). | No invitation from GP (F, P). Not recorded as immunosuppressed in primary-care medical records (F, P). Difficulty booking appointment (F). |
| Introduction of a new vaccine | Confirmation from HCP of novel vaccine technology safety. No reports of side-effects in immunosuppressed (C). | Rapid development of vaccine (C). Type of vaccine (e.g. mRNA) and suitability for IMID (C). |
| Mode of administration | None | Needle phobia |
| Reliability and source of vaccine supply | Good availability of appointments (F) | Limited supply during COVID-19 pandemic (F) |
| Risk benefit | Confidence in vaccine safety (C, F, P). Benefits outweigh risk of vaccination-specific side effects (C, F, P). | Risk of side-effects considered greater burden than benefit of vaccination (F). Safety of taking another vaccine close to COVID-19 (F, P). |
| Strength of recommendation | HCP advising vaccination due to IMID treatment (F, P). | No advice from HCP to be vaccinated (F, P). |

C = COVID-19, F = seasonal-flu, P = pneumonia. [a]Newly identified from interviews.

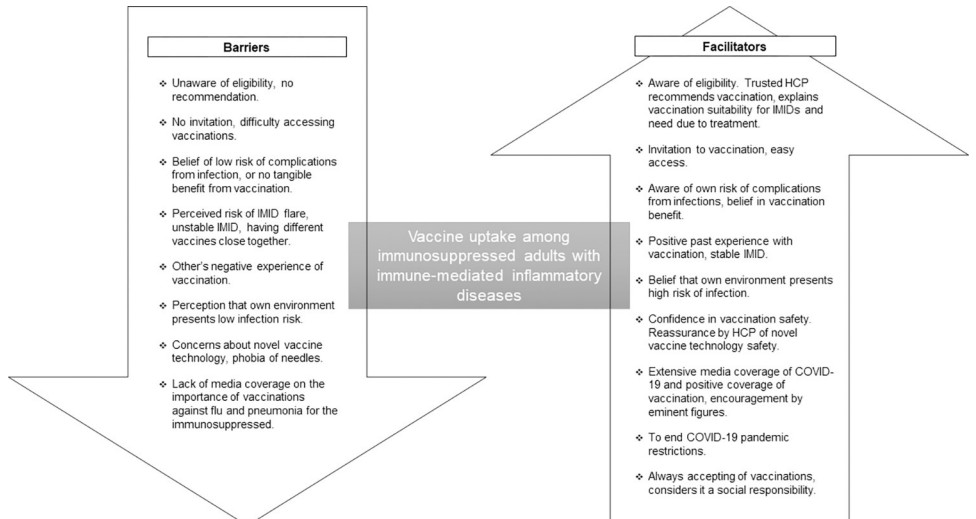

**Fig 1. Diagrammatic summary of the barriers and facilitators of vaccine uptake in immunosuppressed adults with immune-mediated inflammatory diseases.**

## Contextual influences

**Facilitators.** Participants noted that regularly seeing when vaccines against influenza were available in the news and in advertisements at GP surgeries and pharmacies enhanced their awareness of vaccination against influenza. This prompted some to be vaccinated annually (Table 2 and 2A).

The volume of news about the threat and seriousness of COVID-19, benefit of vaccination, and encouragement from government, healthcare professionals (HCPs) and notable scientists

**Table 2. Contextual influences supporting quotes by participant (with participant number and engagement with vaccination).**

Facilitators

 a. It's quite helpful seeing the pop-ups and the ads about flu, because I have a tendency to forget that one, and there's always a bit of difficulty in actually getting an appointment in with a pharmacy or a GP or whatever to actually get that one. And it reminds me, "Oh, it's that time of year, it's out now. I ought to be doing something about it." OPI022, sometimes vaccinated

 b. The scale of it, yeah, and the fact that normally you don't have access to what ICU consultants think or say but a lot of them have had enough so they are going on social media and saying, "Please get vaccinated, we can't cope anymore." We see ambulances piled up, so this is no joke, is it? OPI001, always vaccinated

 c. There's also the Psoriasis UK, I think it is, website, where they had a big section about it, really helped. It made me feel more at ease and that I was okay to get it. OPI162, often not vaccinated

 d. I mean I was trapped in the house. . .It had a complete change on everything, and it was so dominant that the thought then of not taking any help to change that would just seem bizarre. OPI225, always vaccinated

 e. I suppose I didn't ever consider the severity of disease that I might get until they started talking about, I think it was Crohn's and Colitis UK that basically pushed forward saying that people with Crohn's are going to be at more risk because of their immune systems and that kind of got me thinking what else should I get protected against? OPI146, often not vaccinated

Barriers

 f. Probably if I saw targeted ads for the pneumococcal one, it would probably remind me to nag a GP about it again. But there just isn't that much about it. OPI022, sometimes vaccinated

 g. It's pretty much all age-related and never really mentioned immune-suppressed or anything like that. OPI146, often not vaccinated

 h. Because even though flu's around all the time and every year you sort of don't hear about it to the degree of what Covid has, and I guess that that made me feel that it would be more worthwhile to get it. OPI129, often not vaccinated

primed most to get the SARS-CoV-2 vaccine (2b). Seeing the impact of the vaccination programme on hospitalisation and deaths validated its worth.

Many participants sought out information about vaccination from patient organisations, trusted health websites (e.g. NHS) or academic research, and avoided social media. Advice and information provided by their specific IMID patient charity reassured them and endorsed the importance of accepting vaccines, particularly against COVID-19 (2c).

As most shielded during the pandemic, vaccination gave participants the opportunity to see family and not be house-bound (2d). Even for those who had not shielded, the opportunity to re-enter society facilitated vaccination. Being categorised as Clinically Extremely Vulnerable prompted a small number of 'often not vaccinated' participants to consider their infection risk and uptake of other vaccinations. For most, the COVID-19 pandemic affirmed but did not change their belief in the importance of vaccination (2e).

**Barriers.**   Some noted a lack of advertisements for vaccination against pneumococcal pneumonia compared to influenza (2f) may have limited their awareness. Additionally, many reported influenza and pneumococcal vaccination advertisements referred to age eligibility but not immunosuppressed status, so didn't consider them necessary or ask their GP about it (2g). Participants noted how the need to be vaccinated against influenza and pneumococcal pneumonia is not communicated in the media in the same way as COVID-19, which may lessen their perceived importance (2h).

## Individual and group influences

**Facilitators.**   A key facilitator in accepting vaccines against COVID-19, influenza and pneumococcal pneumonia was a higher perceived susceptibility to infection due to being immunosuppressed and the benefit of vaccination to prevent hospitalisation and death (Table 3 and 3A). Many believed in vaccination to provide good protection against infection. Although some questioned vaccine efficacy in immunosuppressed people, the expectation of some rather than no protection was enough. Some had the motivation to avoid further illness given the existing burden of their IMID (3b), and several linked a reduction in having influenza and common-colds to the flu-vaccine, facilitating continued vaccination (3c).

Those who had not been vaccinated against influenza and/or pneumococcal pneumonia but had been vaccinated against COVID-19 often perceived the latter as more serious with a greater chance of getting very unwell (3d). As the potential complications from contracting COVID-19 were less well understood it was perceived as having greater potential to threaten their health, particularly with few treatment options. Many felt a great desire to have a COVID-19 vaccination, having waited for it to become available. Vaccination against COVID-19 was also seen as a social responsibility to protect others and minimise NHS burden (3e). A few had taken the vaccine against influenza to protect elderly family members.

Many participants with additional risk factors for vaccination felt these were important reasons for being vaccinated against flu and/or pneumonia, in some cases above and beyond being immunosuppressed. Working in an environment considered high risk, such as a school or hospital, also facilitated acceptance of vaccination against these infections. A positive experience with previous vaccinations facilitated continued engagement with vaccination against influenza and COVID-19. Some looked to online forums to see the COVID-19 vaccination experiences of other immunosuppressed-IMID patients. Seeing side-effects were unrelated to their IMID contributed to acceptance (3f).

Vaccine acceptance was also greatly facilitated by having trust in the HCP providing the recommendation, in particular it coming from a nurse, GP or specialist who treated or was familiar with their IMID. Some participants also said confirmation from HCPs that

**Table 3. Individual and group influences supporting quotes, by participant (with participant number and engagement with vaccination).**

Facilitators

 a. *I can't fight diseases properly because I'm immune supressed, and so that runs all sorts of risks that things could get a lot worse, and flu can be potentially dangerous, and so for me it's a risk that obviously I don't want to take, or put any burden, extra burden on the NHS if I don't have to. So that's why, for me, being vaccinated is important.* OPI019, always vaccinated

 b. *I've been diagnosed for four years, and I haven't had long-term remission in that time. So I will do anything in order to make sure that things aren't worse, don't get worse. So the thought of as I've said, I'll be proactive because I don't want to get ill, I don't want to have anything else.* OPI225, sometimes vaccinated

 c. *Ever since I've had the flu vaccine, whether it's helped or not I don't know but I've not seemed to suffer with colds. . . I've not been ill with anything as bad as I was prior to taking vaccines and stuff. So for me it's just an ongoing thing I'll carry on doing every year.* OPI206, always vaccinated

 d. *The flu, I can't say it's not serious. It is serious but it's less than the COVID. So, for example, if we give an example, one of every one hundred people will die of COVID, but one out of 1,000, 2,000 people will die from the flu. So if you compare the researches there, I think that taking the COVID vaccine will be more important than flu vaccine.* OPI210, often not vaccinated

 e. *I believed in it and do believe in it from a societal perspective and a societal benefit of controlling the illness.* OPI058 sometimes vaccinated

 f. *And as I say others with the same disease that were obviously older than me that had already had the vaccines and to be fair there's not too many of them had any real side-effects.* OPI003, often not vaccinated

 g. *I basically asked him can I have it with my Infliximab and he said, "Yes, because it's not a live vaccine, it won't interact with that, it won't interact with your gut." So yeah, generally he alleviated any issues I had. . .You've got to be within the therapeutic level for it to reduce the inflammation and I was just worried it was going to interact with that and it was going to drop below the level and my Crohn's would get worse.* OPI146, often not vaccinated

Barriers

 h. *I hardly see my GP, I don't really. . . I mean if they recommended it for Crohn's I'd want my Crohn's people to tell me that that's true.* OPI225, sometimes vaccinated

 i. *Maybe for me it was my lack of understanding how seriously immunosuppressed I was or I am, that it didn't occur to me that there would be a benefit.* OPI129, often not vaccinated

 j. It is no longer something I am even contemplating based on how long my problems went on and the fact that I'm still not fully over them. OPI058, sometimes vaccinated

 k. *I think time and other health issues push it down. Even though it's a high priority to cover yourself, it's not been a high priority among everything else that's been going on, if that makes sense.* OPI042, sometimes vaccinated

vaccination would not trigger an IMID flare or reduce their medication's effectiveness had been important (3g), and some 'sometimes' and 'often not' vaccinated patients said such information would be important when deciding whether to accept a vaccine.

**Barriers.** A key barrier to vaccination against influenza and pneumonia was not being aware these were recommended for immunosuppressed-IMID patients. This was a particularly common reason for being vaccinated against influenza but not pneumonia among the 'sometimes' vaccinated participants. In many cases, the recommendation to be vaccinated against influenza and/or pneumonia came over a year after commencing immunosuppressing medication, delaying vaccination. For some, a vaccine recommendation from a HCP unfamiliar with their IMID was insufficient grounds to accept it (3h).

Low perceived risk of infection despite being immunosuppressed and feeling no tangible benefit from vaccination against influenza and/or pneumonia (3i) was another barrier for participants. A small number of people felt this was because they were not in an at-risk age category or were in an environment where they were at low risk of contracting these infections e.g. shielding, retired.

Another barrier for some participants was the belief that an IMID flare was triggered by a specific vaccine. After considering their risk of infection, one participant recommended flu vaccination, but others rejected further doses of a vaccine as the impact of a flare was considered greater than the potential benefits of vaccination (3j). These participants found it difficult to get support from HCPs, who either dismissed concerns or could only provide general recommendations. One participant felt they could not be persuaded even if given reassurances

that the risk of flaring again was low, whereas others felt IMID-specific advice and/or the opportunity to discuss the risks of flaring again may help to overcome their fear. Despite these concerns, perceived risk of a flare following vaccination did not rule out accepting a different vaccine. Having family/friends who believed the influenza vaccine triggers flu-like symptoms also put some off from taking it.

Being busy with other appointments for their IMID lowered the priority of vaccination for some (3k). When their IMID was not fully under control, either from ongoing symptoms or managing new medications, vaccination was delayed.

## Vaccine or vaccination specific issues

**Facilitators.** Direct invitation from their GP surgery was the prompt for many to start getting vaccinated against flu and pneumonia. Yearly invitations reminded many to continue having the flu vaccine (Table 4 and 4A). These were mostly received by participants with at-risk factors for infection, although a couple without also reported receiving invitations. Some proactively booked vaccinations themselves, being aware of needing them. Ease of getting an appointment through their GP surgery, or ability to be vaccinated at a private pharmacy in the absence of GP appointments, facilitated flu vaccination.

Participants reported that the design of the mass vaccination programme against COVID-19 –repeated direct invitation, good appointment availability, active checking of vaccination status–removed barriers experienced with other vaccines (4b).

HCPs emphasising their vulnerability to infection alongside a clear recommendation to be vaccinated was a key reason for accepting particular vaccinations. Those who had not been vaccinated against influenza and/or pneumonia said this would be a strong encouragement (4c). Some suggested repeated messaging did or would facilitate getting vaccinated (4d). A

**Table 4. Vaccine or vaccination specific issues supporting quotes, by participant (with participant number and engagement with vaccination).**

Facilitators

 a. *Every year they contact me to get me booked in*. OPI019, always vaccinated

 b. *The main thing I find is, with the COVID vaccine it seems GP surgeries were actively checking that their patients were booked in and had had it. Whereas I feel like–last year for example I had my [flu] vaccine done privately and I don't think they told my GP about it. Nobody bothers to call me, check up on me. It feels like in comparison nobody really cares. . . potentially there are some things from there that they could apply to the pneumonia and the flu vaccine.* OPI022, sometimes vaccinated

 c. *I think the only thing that would make me get it at this moment in time is a healthcare professional telling me that I should because of the situation I'm in.* OPI162, often not vaccinated

 d. *I suppose if it's repetition of reminding you that you should have it, like even having this conversation makes me think why am I not having it?* OPI135, often not vaccinated

 e. *If I said no I wouldn't of got the immunosuppressants and then I would have become even more ill. So I had no choice, I was pushed in the corner, that's why I had to have it . . . I wanted the lupus flares to stop.* OPI089, sometimes vaccinated

 f. *Any reaction you get is far less than the risk of not having the injection.* OPI035, sometimes vaccinated

 g. As the vaccination programme expanded, my confidence in that expanded. OPI019, always vaccinated

Barriers

 h. *GP has said, "You're immunosuppressed, so you should be being invited by the system if you haven't had one." And Rheumatology has said the same as well. When I've said, "I've never been invited for pneumococcal," they say, "Well, your GP should be sorting that out for you." And GP says, "Well, if you're tagged correctly in the system you'll get invited." I guess they view it as an admin task. . . It's more the inconvenience of having to push for it. The costs outweighs the benefits for me at the moment on that.* OPI022, sometimes vaccinated

 i. *What I tried to do was go back and look at some scientific information, and again just weighed up the risks and thought it was worth the risk.* OPI035, sometimes vaccinated

 j. *As soon as I had it I had really bad side effect which makes me cough for nearly, I can say, 20 days, even I think more. It was like so bad coughing. . . there were a fever and felt unwell. . . So I am not going to have this again if I have the same symptoms or same side effects.* OPI210, often not vaccinated

couple of participants who accepted influenza and/or pneumonia vaccines had felt coerced into this decision (4e).

Overall confidence in vaccination safety was high, more so for flu and pneumonia as they had been available for longer. Any risks were perceived as minor e.g. sore arm, compared to the risk of not being vaccinated (4f). Observing the COVID-19 vaccination programme roll-out without any reported consequences to immunosuppressed-IMID patients boosted confidence in accepting the vaccine (4g).

**Barriers.**   For many, key barriers to being vaccinated against influenza and/or pneumo-coccal pneumonia were not receiving an invitation from their GP or recommendation from a HCP to get vaccinated. Many 'sometimes vaccinated' participants received clear recommendations and direct invitations to have their influenza but not pneumonia vaccination. Some had been advised to take infection control measures but not received advice on vaccines by a specialist or pharmacist administering the medication.

Some who were aware of their vaccine eligibility reported difficulties when trying to rectify the reasons for their lack of invitation e.g. not recorded as immunosuppressed, with their GP practice (4h). COVID-19 vaccines being new and rapidly developed had concerned some, including suitability of mRNA vaccines for their IMID and reports of side-effects in media, but HCP reassurances and/or perceived benefits of being vaccinated overcame this (4i).

Some chose to not take influenza vaccination following side-effects (including swelling and flu symptoms). The impact of side-effects were perceived to be greater than the potential benefits of vaccination (4j). Some participants expressed concerns about the safety of having several vaccinations close together, and prioritised COVID-19 over influenza or pneumonia. A fear of needles was an additional barrier for one participant.

## Discussion

While several quantitative studies have been published on the topic of vaccine hesitancy in people with IMIDs, qualitative studies which give greater insight into the thought process behind such decision-making are few. This is the first qualitative study to explore barriers and facilitators to vaccination against COVID-19 among immunosuppressed adults with IMIDs. It was conducted in winter 2021, when all adults in the UK had been offered at-least two doses of COVID-19 vaccine and the booster programme had begun. It found that determinants of engagement with vaccination are similar between people with different IMIDs, and among those treated with conventional or biologic immune-suppressing drugs. Our findings align with the SAGE working group vaccine hesitancy matrix and a previous qualitative research study on vaccine hesitancy about vaccination against pneumococcal and seasonal flu vaccination in people with RA [21], with barriers and facilitators spanning contextual, individual/group and vaccine/vaccination specific influences. Similar to other studies qualitatively exploring vaccine hesitancy in select population groups [22,23], we identified additional factors influencing the decision to be vaccinated that were specific to this patient group, such as concerns of vaccination inducing IMID-flare, as well as those specific to the COVID-19 pandemic, such as vaccination as an opportunity to end social restrictions. We also demonstrated how some barriers and facilitators differ from one vaccine to another.

As the focus on COVID-19 and its risks lessen, the perceived importance of vaccination against it may reduce to a level similar to that against other vaccine-preventable illnesses. Thus, it is important to be mindful of the current barriers to vaccination against influenza and pneumococcal pneumonia, and to implement strategies to mitigate against vaccine hesitancy in future vaccinations against COVID-19.

Participants' understanding of increased risk of infection and complications from infections was an important reason for getting vaccinated in our study as reported previously [21,24–27]. Similarly, lack of awareness about eligibility for getting vaccinated has been reported as a barrier to vaccination [21], but was particularly prominent for pneumococcal vaccine uptake in our study.

Our findings suggest that there is an unmet need to improve awareness of eligibility for vaccination. HCP recommendation and annual vaccination reviews improve vaccine uptake, potentially because physician recommendation overcomes ambivalence to vaccination [24,28–31]. Interestingly, we found that many participants relied upon and sought advice on the safety and suitability of vaccination with respect to their disease or its treatment from their specialist, despite vaccination being a core activity of primary care. It perhaps highlights a bias among patients perceiving their specialists' opinion being more trustworthy in this context. In clinical practice, primary care physicians play an important role in optimising vaccine awareness and receptiveness, for example as part of annual health checks. Therefore, we suggest that vaccination history should be reviewed during IMID consultations in both primary and secondary care, and reasons for non-vaccination explored and addressed in both settings. Such enquiries could be easily incorporated in annual reviews of chronic diseases. In those not up to date with vaccination, their primary-care provider should be informed of the need of vaccination. This is particularly important for patients on hospital-prescribed biologics as primary-care records may not include this information, precluding direct invitations. Additionally, targeted invitation for vaccination should be offered to patients on long-term immunosuppression [18]. Highlighting this as a key eligibility criteria for vaccinations on posters/adverts aimed at the general population may also increase awareness.

As reported elsewhere, we found concern regarding vaccination-associated IMID flares emerged as a potential barrier to vaccination [21,30,32]. This dissipated following reassurance from a trusted HCP. Thus, HCPs should explore and address any such concerns when checking vaccination status.

IMID patient organisations were identified as a useful and trusted source of information in participants' decision-making process for vaccination in our study, which is consistent with recent survey [31]. Patients actively avoided taking vaccine information from social media, contrary to vaccine-hesitant individuals in the general public [23,33]. This highlights that different types of media may be viewed differently by IMID-patients when making vaccination decisions, which may be due to the IMID-specific considerations they have and the availability of IMID-specific information from patient charities. It is important therefore that patient organisations and charities continue to provide clear advice and address patient concerns about vaccinations on their websites.

Although some participants in this study reported concerns regarding the rapid development of the COVID-19 vaccines and suitability of mRNA vaccines, these did not ultimately act as a barrier to vaccination. In another study, patients most concerned about new vaccine technology and the lack of long-term data [34] were more likely to be unsure of or decline vaccination, suggesting that this is still an important concern and should be addressed where appropriate.

Before the COVID-19 pandemic, a Canadian study explored factors influencing influenza and pneumococcal pneumonia vaccine uptake among RA patients [21]. It highlighted multiple barriers identified in the present work, but also reported distrust of health systems and pharmaceuticals as key barriers. These latter issues were not identified as barriers by us, potentially reflecting greater trust in the NHS and may have been influenced by the health benefits seen from vaccines against COVID-19. The present study was conducted during the COVID-19 pandemic, when several scientific, government, and public figures in the UK actively

promoted the safety and efficacy of vaccination against COVID-19. Vaccination against COVID-19 was delivered through the NHS vaccination program at no cost to the patient at the point of delivery. Consequently, vaccines may not have been perceived as being produced for benefit by pharmaceutical companies in the present study.

Strengths of this study include broad eligibility criteria across several IMIDs and nation-wide recruitment. Participants had a wide age range, varying degrees of engagement with vaccination, and there was adequate participation of non-white participants. These factors enhance the transferability of our findings. Although caution should be taken when interpreting the results with a small sample size, thematic data saturation was observed. A non-clinician interviewer reduced the risk of response bias and the involvement of a second coder and clinician enhanced the rigour of analysis. Limitations include that participants were a self-selecting group who may be more engaged in their care and a lack of participants who had not accepted a vaccination against COVID-19, however, our purposive sampling ensured a balanced mix of engagement with other vaccinations.

In conclusion, the determinants of vaccination against COVID-19, influenza and pneumo-coccal pneumonia among immunosuppressed-IMID patients are numerous and multifaceted, and vary by vaccine-type. Many barriers could be addressed in clinical practice and through public-health messaging, although further research is needed to determine the effectiveness of such measures.

## Supporting information

**S1 Table. Participant characteristics.**
(DOCX)

**S1 Methods. Participant eligibility criteria and recruitment.**
(DOCX)

**S2 Methods. Interview guide questions.**
(DOCX)

## Acknowledgments

We would like to extend our thanks to the following charities who supported with recruitment: National Rheumatoid Arthritis Society, National Ankylosing Spondylitis Society, Crohns and Colitis UK, Vasculitis UK, Lupus UK, and Psoriasis Association UK.

## Author Contributions

**Conceptualization:** Amy Fuller, Jennie Hancox, Kavita Vedhara, Tim Card, Christian Mallen, Jonathan S. Nguyen Van-Tam, Abhishek Abhishek.

**Formal analysis:** Amy Fuller, Jennie Hancox.

**Funding acquisition:** Abhishek Abhishek.

**Investigation:** Amy Fuller.

**Project administration:** Amy Fuller.

**Resources:** Amy Fuller.

**Supervision:** Abhishek Abhishek.

**Validation:** Jennie Hancox.

**Visualization:** Amy Fuller, Jennie Hancox.

**Writing – original draft:** Amy Fuller.

**Writing – review & editing:** Jennie Hancox, Kavita Vedhara, Tim Card, Christian Mallen, Jonathan S. Nguyen Van-Tam, Abhishek Abhishek.

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
