## [Decision Letter · Decision Letter 0]

16 Jun 2022

PONE-D-22-10935Barriers and facilitators to vaccination uptake against COVID-19, influenza, and pneumococcal pneumonia in immunosuppressed adults with immune-mediated inflammatory diseases: a qualitative interview study during the COVID-19 pandemicPLOS ONE

Dear Dr. Fuller,

Thank you for submitting your manuscript to PLOS ONE. After careful consideration, we feel that it has merit but does not fully meet PLOS ONE’s publication criteria as it currently stands. Therefore, we invite you to submit a revised version of the manuscript that addresses the points raised during the review process.

We look forward to receiving your revised manuscript.

Kind regards,

Latika Gupta

Academic Editor

PLOS ONE

Journal Requirements:

"I have read the journal's policy and the authors of this manuscript have the following competing interests:

AF, JH, KV, TC and JSN-V-T declare no competing interests.

CM declares grants awarded from MRC, AHRC BMS and Versus Arthritis, and is Director of the NIHR SPCR.

AA declares grants from AstraZeneca and Oxford Immunotec, personal payments from UpToDate, Springer, Menarini and Cadilla pharmaceuticals, consulting fees from Inflazome and NGM Biopharmaceuticals, meeting attendance/travel payments from Pfizer, and is co-chair for the OMERACT CPPD classification criteria and ACR/EULAR CPPD classification criteria working groups.

JSN-V-T was seconded to the Department of Health and Social Care, England (DHSC) until 31st March 2022. The views expressed in this manuscript are those of its authors and not necessarily those of DHSC or the Joint Committee on Vaccination and Immunisation (JCVI)."

Additional Editor Comments:

There manuscript is well written and timely. Minor edits may be considered.

Reviewers' comments:

Reviewer's Responses to Questions

**Comments to the Author**

1. Is the manuscript technically sound, and do the data support the conclusions?

Reviewer #1: Yes

Reviewer #2: Yes

2. Has the statistical analysis been performed appropriately and rigorously? 

Reviewer #1: Yes

Reviewer #2: Yes

3. Have the authors made all data underlying the findings in their manuscript fully available?

Reviewer #1: Yes

Reviewer #2: Yes

4. Is the manuscript presented in an intelligible fashion and written in standard English?

Reviewer #1: Yes

Reviewer #2: Yes

5. Review Comments to the Author

Reviewer #1: The authors conducted a qualitative study to explore barriers and facilitators to vaccination against COVID-19 among immunosuppressed adults with IMIDs in UK. It found that determinants of engagement with vaccination are similar between people with different IMIDs with barriers and facilitators spanning contextual, individual/group and vaccine/vaccination specific influences.

The authors could redesign the data visualization of the facilitator/barrier tables to make them more impactful.

Reviewer #2: The manuscript is well written and well-timed as even in the midst of the pandemic vaccine refusals have been rampant and unfortunately so even in the immunocompromised group.

Multiple quantitative studies have been published on the topic of vaccine hesitancy but qualitative studies are rarer, especially in the patient population with inflammatory immune-mediated diseases. Gives more insight in the thought process of the actual patients.

It is interesting that patients actually prefer using patient resource centers and dedicated websites for vaccine information over social media websites. At the same time advertisements at GP offices actually encouraged vaccination uptake. So it’s the different type of media that is viewed differently by patients when it comes to making informed health decisions.

This study brings out a fact that patients with chronic disease tend to rely on their specialists even for basic primary care like vaccination and there appears to be an underlying bias that specialists’ opinion about vaccination is more trustworthy. While in clinical practice primary care physicians can play an important role in increasing vaccine awareness and receptiveness. Yearly checklists of things to be done for health maintenance are one of the major work areas of primary care.

It is interesting to note that there is a difference from the Canadian study where there was mistrust about pharmaceutical companies which was not seen in this UK-based study. It opens an arena about are their differences in vaccine uptake and vaccine hesitancy.

Discussing the above points in a little more detail might improve the manuscript even more. Overall, a well written and thought about paper

6. PLOS authors have the option to publish the peer review history of their article (what does this mean?). If published, this will include your full peer review and any attached files.

Reviewer #1: No

Reviewer #2: **Yes: **Tulika chatterjee

---

## [Author Response · Author response to Decision Letter 0]

27 Jul 2022

Reviewer 1

Comment: The authors could redesign the data visualization of the facilitator/barrier tables to make them more impactful.

Action: We agree that an impactful visual of the data would benefit this manuscript. We feel that it is important to retain Table 1 as it shows how the data maps to sub-categories of the SAGE matrix by vaccine-type, so have kept this within the manuscript. We have created a diagrammatic summary (Fig 1) of the barriers and facilitators, providing the reader with a quick and impactful overview of the data. The figure caption can be found on page 10, lines 151 – 152.

We made a few minor amendments in Table 1 to make the data included within it clearer. 

Reviewer 2

Reviewer 2 noted that discussing the following points in a little more detail might improve the manuscript even more. 

Comment: Multiple quantitative studies have been published on the topic of vaccine hesitancy but qualitative studies are rarer, especially in the patient population with inflammatory immune-mediated diseases. Gives more insight in the thought process of the actual patients.

Action: Thank you, we agree and have added comment to this point on page 20, lines 305 – 307 and lines 315 – 323.

Comment: It is interesting that patients actually prefer using patient resource centers and dedicated websites for vaccine information over social media websites. At the same time advertisements at GP offices actually encouraged vaccination uptake. So it’s the different type of media that is viewed differently by patients when it comes to making informed health decisions.

Action: This is a great point, especially when social media has been found to be a source of information and misinformation among vaccine-hesitant individuals in the general population. We have added discussion to this point on page 22, lines 363 – 370.

Comment: This study brings out a fact that patients with chronic disease tend to rely on their specialists even for basic primary care like vaccination and there appears to be an underlying bias that specialists’ opinion about vaccination is more trustworthy. While in clinical practice primary care physicians can play an important role in increasing vaccine awareness and receptiveness. Yearly checklists of things to be done for health maintenance are one of the major work areas of primary care.

Action: Thank you for raising this. We have added discussion relating to this on page 21, lines 339 – 351.

Comment: It is interesting to note that there is a difference from the Canadian study where there was mistrust about pharmaceutical companies which was not seen in this UK-based study. It opens an arena about are their differences in vaccine uptake and vaccine hesitancy.

Action: We have added further discussion to this point on page 23, lines 383 – 388.

Editorial requests from PLOS ONE

In addition, we have addressed the following requests from the journal.

Action: The manuscript and supporting information documents have been edited to meet the PLOS ONE style requirements.

2. Please provide additional details regarding participant consent. In the ethics statement in the Methods and online submission information, please ensure that you have specified what type you obtained (for instance, written or verbal, and if verbal, how it was documented and witnessed). 

Action: Participants provided their consent using an online consent forms via Microsoft Teams. We have added this detail into the ‘Ethical approval’ statement in the Methods (page 6, lines 87 – 88) and online submission information.

3. Thank you for stating the following in the Competing Interests section. Please confirm that this does not alter your adherence to all PLOS ONE policies on sharing data and materials, by including the following statement: "This does not alter our adherence to PLOS ONE policies on sharing data and materials.” 

Action: We have added the required statement to our Competing Interests statement which is written in full below: 

“I have read the journal's policy and the authors of this manuscript have the following competing interests:

AF, JH, KV, TC and JSN-V-T declare no competing interests.

CM declares grants awarded from MRC, AHRC BMS and Versus Arthritis, and is Director of the NIHR SPCR.

AA declares grants from AstraZeneca and Oxford Immunotec, personal payments from UpToDate, Springer, Menarini and Cadilla pharmaceuticals, consulting fees from Inflazome and NGM Biopharmaceuticals, meeting attendance/travel payments from Pfizer, and is co-chair for the OMERACT CPPD classification criteria and ACR/EULAR CPPD classification criteria working groups.

JSN-V-T was seconded to the Department of Health and Social Care, England (DHSC) until 31st March 2022. The views expressed in this manuscript are those of its authors and not necessarily those of DHSC or the Joint Committee on Vaccination and Immunisation (JCVI).

This does not alter our adherence to PLOS ONE policies on sharing data and materials.”

4. In your Data Availability statement, you have not specified where the minimal data set underlying the results described in your manuscript can be found. PLOS defines a study's minimal data set as the underlying data used to reach the conclusions drawn in the manuscript and any additional data required to replicate the reported study findings in their entirety. All PLOS journals require that the minimal data set be made fully available. 

Action: We apologise that our previous Data Availability statement did not set out where the minimal dataset can be found. As per the guidance on your website for qualitative data https://journals.plos.org/plosone/s/data-availability#loc-unacceptable-data-access-restrictions, transcript excerpts relevant to the study are included within the paper. The interview guide has been included as a supplementary file. Additional qualitative data and audio recordings are archived in the University of Nottingham servers using password protection. For requests to access this additional data the Chief Investigator can be contacted abhishek.abhishek@nottingham.ac.uk.

5. PLOS requires an ORCID iD for the corresponding author in Editorial Manager on papers submitted after December 6th, 2016. Please ensure that you have an ORCID iD and that it is validated in Editorial Manager. 

Action: The ORCID ID for the corresponding author has now been added and validated in Editorial Manager.

6. Please include captions for your Supporting Information files at the end of your manuscript, and update any in-text citations to match accordingly. 

Action: Captions for the Supporting Information files have now been provided at the end of the manuscript with in-text citations to match.

7. Please review your reference list to ensure that it is complete and correct. 

Action: We can confirm the reference list is complete and correct, and we have ensured it meets the PLOS ONE reference formatting requirements.

---

## [Decision Letter · Decision Letter 1]

12 Aug 2022

Barriers and facilitators to vaccination uptake against COVID-19, influenza, and pneumococcal pneumonia in immunosuppressed adults with immune-mediated inflammatory diseases: a qualitative interview study during the COVID-19 pandemic

PONE-D-22-10935R1

Dear Dr. Fuller

We’re pleased to inform you that your manuscript has been judged scientifically suitable for publication and will be formally accepted for publication once it meets all outstanding technical requirements.

Kind regards,

Latika Gupta

Academic Editor

PLOS ONE

Additional Editor Comments (optional):

All concerns have been adequately addressed and the revision is acceptable for publication.

Reviewers' comments:

Reviewer's Responses to Questions

**Comments to the Author**

1. If the authors have adequately addressed your comments raised in a previous round of review and you feel that this manuscript is now acceptable for publication, you may indicate that here to bypass the “Comments to the Author” section, enter your conflict of interest statement in the “Confidential to Editor” section, and submit your "Accept" recommendation.

Reviewer #1: All comments have been addressed

2. Is the manuscript technically sound, and do the data support the conclusions?

Reviewer #1: Yes

3. Has the statistical analysis been performed appropriately and rigorously? 

Reviewer #1: Yes

4. Have the authors made all data underlying the findings in their manuscript fully available?

Reviewer #1: Yes

5. Is the manuscript presented in an intelligible fashion and written in standard English?

Reviewer #1: Yes

6. Review Comments to the Author

Reviewer #1: The aim of this study is to explore barriers and facilitators to COVID-19, influenza, and pneumococcal vaccine

uptake in immunosuppressed adults with immune-mediated inflammatory diseases. The method used a qualitative analysis

of a semi-structured interview between November 2021 and January 2022.

The authors have responded fully to the reviewers' comments.

7. PLOS authors have the option to publish the peer review history of their article (what does this mean?). If published, this will include your full peer review and any attached files.

Reviewer #1: No

---

## [Editor Report · Acceptance letter]

30 Aug 2022

PONE-D-22-10935R1 

Barriers and facilitators to vaccination uptake against COVID-19, influenza, and pneumococcal pneumonia in immunosuppressed adults with immune-mediated inflammatory diseases: a qualitative interview study during the COVID-19 pandemic 

Dear Dr. Fuller:

I'm pleased to inform you that your manuscript has been deemed suitable for publication in PLOS ONE. Congratulations! Your manuscript is now with our production department. 

Kind regards, 

on behalf of

Dr. Latika Gupta 

Academic Editor

PLOS ONE